# Development of Lymphopenia during Therapy with Immune Checkpoint Inhibitors Is Associated with Poor Outcome in Metastatic Cutaneous Melanoma

**DOI:** 10.3390/cancers14133282

**Published:** 2022-07-05

**Authors:** Dirk Tomsitz, Max Schlaak, Sarah Zierold, Giulia Pesch, Thomas U. Schulz, Genoveva Müller, Christine Zecha, Lars E. French, Lucie Heinzerling

**Affiliations:** 1Department of Dermatology and Allergy, University Hospital, Ludwig Maximilian University of Munich, 80337 Munich, Germany; sarah.zierold@med.uni-muenchen.de (S.Z.); giulia.pesch@med.uni-muenchen.de (G.P.); thomas.schulz@med.uni-muenchen.de (T.U.S.); genoveva.mueller@med.uni-muenchen.de (G.M.); christine.zecha@med.uni-muenchen.de (C.Z.); lars.french@med.uni-muenchen.de (L.E.F.); lucie.heinzerling@med.uni-muenchen.de (L.H.); 2Department of Dermatology, Venereology and Allergology, Charité—Universitätsmedizin Berlin, Corporate Member of Freie Universität Berlin and Humboldt-Universität zu Berlin, 10117 Berlin, Germany; max.schlaak@charite.de; 3Dr. Philip Frost Department of Dermatology & Cutaneous Surgery, Miller School of Medicine, University of Miami, Miami, FL 33136, USA; 4Department of Dermatology, University Hospital Erlangen, 91054 Erlangen, Germany

**Keywords:** lymphopenia, immune checkpoint inhibitors, melanoma, predictive markers, outcome

## Abstract

**Simple Summary:**

Predictive markers are necessary for immune checkpoint inhibitor (ICI) therapy. The aim of our retrospective study was to investigate the relationship between the occurrence of lymphopenia under ICI and disease outcome. A total of 116 patients with metastatic melanoma who received ICI therapy with normal lymphocyte counts at baseline were analyzed. Lymphopenia occurred in 42.2% of patients with a mean onset after 17 weeks (range 1–180 weeks). The occurrence of lymphopenia during immunotherapy was significantly associated with a shorter PFS and OS. Patients who developed lymphopenia (*n* = 49) had a mean PFS of 13.3 months (range 1–67 months) compared to 16.9 months (range 1–73 months) for patients who did not develop lymphopenia (*n* = 67; *p* = 0.025). Similarly, patients with lymphopenia had a significantly shorter OS of 28.1 months (range 2–70 months) compared with 36.8 months (range 4–106 months) in patients who did not develop lymphopenia (*p* = 0.01).

**Abstract:**

Predictive markers for immune checkpoint inhibitor (ICI) therapy are needed. Thus, baseline blood counts have been investigated as biomarkers, showing that lymphopenia at the start of therapy with (ICI) is associated with a worse outcome in metastatic melanoma. We investigated the relationship between the occurrence of lymphopenia under ICI and disease outcome. Patients with metastatic melanoma who had undergone therapy with ICI were identified in our database. Only patients with a normal lymphocyte count at baseline were included in this retrospective study. Progression-free survival (PFS) and overall survival (OS) were compared between patients in which lymphopenia occurred during ICI therapy and those who did not develop lymphopenia. In total, 116 patients were analyzed. Lymphopenia occurred in 42.2% of patients, with a mean onset after 17 weeks (range 1–180 weeks). The occurrence of lymphopenia during immunotherapy was significantly associated with a shorter PFS and OS. Patients who developed lymphopenia (*n* = 49) had a mean PFS of 13.3 months (range 1–67 months) compared to 16.9 months (range 1–73 months) for patients who did not develop lymphopenia (*n* = 67; *p* = 0.025). Similarly, patients with lymphopenia had a significantly shorter OS of 28.1 months (range 2–70 months) compared with 36.8 months (range 4–106 months) in patients who did not develop lymphopenia (*p* = 0.01). Patients with metastatic melanoma who develop lymphopenia during ICI therapy have a worse prognosis with significantly shorter PFS and OS compared with patients who do not develop lymphopenia.

## 1. Introduction

Immune checkpoint inhibitors (ICI) have significantly improved overall survival in patients with metastatic melanoma. However, even with combined immunotherapy, some patients do not benefit from treatment. Rates for primary resistance range between 40% and 65% for anti-programmed death 1 (PD1) therapy with pembrolizumab or nivolumab and can reach more than 80% for therapy with anti-cytotoxic T-lymphocyte-associated antigen 4 (CTLA4) monoclonal antibody [1,2,3]. In patients who were treated with combined ipilimumab and nivolumab, the primary resistance rate was decreased to 42% [3]. Prognostic factors associated with a worse outcome independent of the chosen therapeutic regime include elevated serum lactate dehydrogenase, NRAS-mutated melanoma, and the presence of brain metastases [4,5,6]. Biomarkers which reliably predict response to ICI are still needed. Baseline biomarkers to select patients who will benefit from therapy have been investigated in various studies. However, many have not been proven to help in decision making. A high relative eosinophil count and a high relative lymphocyte count are associated with better overall survival (OS) in patients treated with ICI [7], while lymphopenia at baseline [8] and the use of antibiotics in the month preceding immunotherapy are associated with worse treatment response and OS in patients treated with ICI [9]. 

Early changes after the start of checkpoint inhibitor therapy have also been assessed. Here, an increase in eosinophils upon the initiation of treatment is positively associated with outcome in melanoma, especially in patients who are treated with ICI. Patients with a percentage of eosinophils greater than 5% showed a median overall survival of 19 months compared to 10 months for patients with a percentage of eosinophils less than 5% [10]. This is in accordance with another retrospective study showing that an eosinophil count of more than 200 in month 1 after start of ICI was associated with increased PFS [8]. In reported studies, the occurrence of lymphopenia after 6 weeks in patients with non-small-cell lung cancer treated with nivolumab and after 3 months in patients with solid cancer treated with anti-PD1 antibodies was also associated with decreased PFS [8,11]. The assessment of lymphocyte counts at these fixed time-points does not, however, take into account the impact of the occurrence of lymphopenia at any other time during ICI therapy. In addition, observations of therapy regimens with combined ipilimumab and nivolumab have not yet been described. 

Lymphocytes play a crucial role in achieving an antitumor immune response [12]. CTLA4 is exclusively expressed on their surface, while PD1 is also expressed on other activated immune cells such as macrophages, dendritic cells, or Langerhans cells. By blocking the immune checkpoints, exhausted lymphocytes are reactivated and regain their anti-tumor function. During the course of ICI therapy, lymphocyte counts can decrease. Severe lymphopenia occurs in 2.8–11% of cases [3,13,14]. Since tumor response is often associated with the T-cell infiltration of metastases, it could be postulated that early lymphopenia is due to migration to the tumor sites [15]. 

In this study, the relationship between the occurrence of lymphopenia at any time under mono- or combined ICI therapy and disease outcome in metastatic melanoma was investigated. 

## 2. Materials and Methods

Between January 2013 and April 2021, all patients with metastatic cutaneous melanoma who were treated with ICI (ipilimumab, nivolumab, pembrolizumab, and/or combined ipilimumab and nivolumab) were identified in our electronic database and screened. Only patients with a normal lymphocyte count at baseline and a follow-up of at least 6 months were included in this retrospective study. No concomitant diseases, medications, or treatments which could possibly influence lymphocyte counts were documented in the patient cohort analyzed.

Continuous data are presented as means or ranges and categorical data are presented as percentages. Continuous variables were compared using unpaired Student’s t-test. Categorial variables were compared using Fisher’s exact test. Progression-free survival (PFS) and overall survival (OS) were compared between patients in which lymphopenia occurred during ICI therapy and those who showed no lymphopenia. Lymphopenia was defined as less than 1000 lymphocytes per microliter of blood. Log-rank tests were performed to compare PFS and OS between groups. *p* values < 0.05 were considered clinically significant. Statistical analyses were conducted with the statistics software SPSS Version 27. 

## 3. Results

### 3.1. Patients’ Characteristics

Patients with metastatic cutaneous melanoma who started treatment with ICI with a normal baseline lymphocyte count were analyzed for PFS and OS. A total of 116 patients was included in this retrospective study. Lymphopenia with a lymphocyte count of less than 1000 per microliter of blood occurred in 42.2% of patients (*n* = 49), whereas no lymphopenia was detected in 57.8% (*n* = 67). There were no patients with co-morbidities and associated treatments (immunosuppressive treatment, chemotherapy) which could have impacted lymphocyte counts. No blood or blood products, G-CSF, or bone marrow transplants were administered to the patients. In the lymphopenia group, the mean age was 60.6 years (range 37–81 years), with 34 male and 15 female patients. The patients without lymphopenia had a mean age of 62.9 years (range 22–90 years), of which 31 were male and 36 were female. ECOG, LDH levels, and NRAS mutations were well balanced in both groups. The patients with lymphopenia included a higher percentage of individuals with M1d at baseline (42.9% versus 28.4%, *p* = 0.117), as well as a higher percentage of BRAF mutation (51.0% versus 32.8%, *p* = 0.057). Patients who developed lymphopenia were more often treated with combined ipilimumab and nivolumab (63.3% versus 41.8%, *p* = 0.025) compared to anti-PD1-antibody monotherapy nivolumab (14.3% versus 22.4%, *p* = 0.341) or pembrolizumab (12.2% versus 29.9%, *p* = 0.026). During the follow-up period, 33 deaths were reported, of which 20 were in the lymphopenia group and 13 were in the no lymphopenia group (Table 1). In addition, PFS and OS of patients was analyzed using the baseline hematologic prognostic markers neutrophil-to-lymphocyte ratio (NLR) and relative eosinophilic count (REC). Patients with NLR ≥ 5 at baseline (*n* = 12) were compared with those with NLR < 5 at baseline (*n* = 104, Table A1) and REC ≥ 1.5% at baseline (*n* = 68) with REC < 1.5% at baseline (*n* = 48, Table A2), accordingly. 

### 3.2. PFS and OS

Lymphopenia occurred in 42.2% of patients (*n* = 49), with a mean onset after 17 weeks (range 1–180 weeks). Patients with lymphopenia had a mean PFS of 13.3 months (range 1–67 months) compared with 16.9 months (range 1–73 months)for patients without lymphopenia (*n* = 67, *p* = 0.025). Similarly, patients with lymphopenia had a shorter OS of 28.1 months (range 2–70 months) compared with 36.8 months (range 4–106 months) for patients without lymphopenia (*p* = 0.01). Patients with a NLR ≥ 5 at baseline had a mean PFS of 7.3 months compared with 16.3 months (*p* = 0.11) and a mean OS 28.8 months compared with 33.6 months (*p* = 0.48) for patients with a NLR < 5 at baseline. Patients with a REC < 1.5 at baseline had a mean PFS of 13.3 months compared with 16.9 months (*p* = 0.188) and a mean OS 31.6 months compared with 34.2 months (*p* = 0.217) for patients with a REC ≥ 1.5 at baseline; Figure 1).

### 3.3. Onset of Lymphopenia (Time to Lymphopenia)

To determine the influence of the time of onset of lymphopenia on PFS and OS in patients with lymphopenia, subgroups were formed: group A (early onset of lymphopenia, defined as first detection of lymphopenia occurring between the beginning of ICI therapy and up to 4 months after, *n* = 16), group B (delayed onset of lymphopenia, defined as the first detection of lymphopenia 4 to 12 months after the beginning of ICI therapy, *n* = 15), and group C (late onset of lymphopenia, defined as the first detection of lymphopenia 12 months after the beginning of ICI therapy, *n* = 15). Groups were formed according to observed patterns of onset of immune-related adverse events in clinical studies [16,17]. Mean PFS was 11.4 months (range 0–50 months) in group A, 17.1 months (range 0–67 months) in group B, and 11.9 months (range 0–44 months) in group C (*p* = 0.727). The mean OS was 24.4 months (range 2–62 months) in group A, 29.9 months (range 3–70 months) in group B, and 30.1 months (range 3–66 months) in group C (*p* = 0.878) (Table 2). 

### 3.4. Antibiotic Treatment

The study also investigated the influence of infection and antibiotic therapy. Within the patient group who developed lymphopenia, 12 cases (24.5%) of infections were documented during ICI therapy, of which 7 (58.3%) led to hospitalization and intravenous antibiotic treatment. Among the patients who did not develop lymphopenia during ICI treatment, only seven infections (10.4%, *p* = 0.73) were registered with two severe courses leading to hospitalization (28.6%, *p* = 0.035). To investigate the impact of antibiotic intake while under treatment with ICI on PFS and OS, patients with lymphopenia and documented antibiotic treatment during ICI therapy (*n* = 12; 24.5%) were compared with patients without documented antibiotic treatment (*n* = 37; 75.5%). Mean PFS was 14.8 months (range 0–67 months) in patients without antibiotic treatment compared to 8.6 months (range 0–50 months) with antibiotic therapy (*p* = 0.976). Mean OS of patients with antibiotic treatment was 33.2 months (range 3–79) compared to 26.5 months (range 2–70 months) in patients without antibiotic treatment (*p* = 0.73). Additionally, in patients without lymphopenia, patients with documented antibiotic treatment (*n* = 8; 11.9%) were compared with patients without documented antibiotic treatment (*n* = 59; 88.1%). Patients with antibiotic treatment and without lymphopenia had a shorter mean PFS of 14.8 months (range 2–54 months) compared to 17.2 months (range 1–73 months) in patients without antibiotic treatment and without lymphopenia (*p* = 0.322). In patients who did not develop lymphopenia, mean OS was 34.3 months (range 4–104 months) in patients with antibiotic treatment and 37.1 (range 5–106 months) in patients without antibiotic treatment (*p* = 0.512) (Table 3). 

## 4. Discussion

This study shows that the occurrence of lymphopenia at any time during treatment with ICI represents a negative prognostic marker in patients with metastatic cutaneous melanoma. A statistically significant association was observed for both a PFS and OS, with a worse outcome seen for patients who developed lymphopenia, with a PFS of 13.3 months, compared to patients without lymphopenia, who had PFS of 16.9 months. Patients with lymphopenia had an OS of 28.1 months compared to 36.8 months in patients who did not develop lymphopenia. Differences in PFS and OS were observed in patients with an early as well as a late onset of lymphopenia after the initiation of immunotherapy. 

Our data are in accordance with data from a retrospective study of a heterogenic group of solid tumors treated with anti-PD1 antibodies, showing that lymphopenia at the initiation of ICI treatment and/or after 3 months was associated with a shorter PFS [8]. It is worth noting that in that study 51% of the patients received prior radiotherapy and 75% prior chemotherapy, which might explain the high rate of 30% lymphopenia seen in the study population before starting ICI therapy [8]. To reduce the influence of prior treatments, only patients with a normal lymphocyte count at ICI onset were included in our study. 

Additionally, in our cohort, patients with an early onset of lymphopenia (<4 months after start of ICI therapy) showed a trend towards a shorter mean OS compared to patients with a delayed (4–12 months after start of ICI therapy) or a late onset (>12 months after start of ICI therapy) of lymphopenia, but the groups were small. In patients with lymphopenia, a higher rate of general infection and a significantly higher rate of severe infection with hospitalization were documented. At the same time, patients of this group were treated twice as often with antibiotics. A large population study in Denmark with 98,344 individuals showed an association between lymphopenia and increased risk of infection and infection-related deaths [18]. As already documented by Pinato et al., antibiotics can be associated with a reduced benefit of ICI therapy [9]. The intake of antibiotics while under therapy with ICI showed a tendency towards a shorter mean PFS and OS in cases where no lymphopenia occurred and a shorter mean PFS in patients with concomitant lymphopenia. 

Our data underline the importance of circulating lymphocytes detectable in the blood count for antitumor response and protection against infections. The reason for the development of lymphopenia remains unclear but potentially includes (i) the loss of lymphocytes as an immune-related hematological side effect [19], (ii) apoptosis triggered by Fas-ligand expressed within the tumor microenvironment, and (iii) apoptosis due to excessive exhaustion. Lymphopenia as a hematological immune-related adverse event would be in line with the fact that the frequency was significantly higher in patients treated with combined immunotherapy with ipilimumab and nivolumab, which is known to be associated with higher rates of severe immune-related adverse events compared to treatment with anti-PD1- or anti-CTLA-4 antibodies alone [3]. On the other hand, therapy with the anti CTLA-4 monoclonal antibody tremelimumab was shown to restore CD4+ and CD8+ T-cells in 7 of 10 patients with pretreated advanced melanoma and severe lymphopenia [20]. 

The mechanism of apoptosis triggered by Fas-ligand was shown in a genetically engineered melanoma mouse model resistant to checkpoint blockade in which lymphocytes in the periphery were consecutively depleted [21]. Lastly, persistent antigen stimulation and immunosuppressive cytokines lead to T-cell dysfunction with the elevated and sustained expression of inhibitory receptors [22], which is further intensified by ICI.

The worse outcomes seen in patients who develop lymphopenia could be due to changes in the tumor microenvironment. The accumulation of myeloid-derived suppressor cells, type-2 macrophages, or regulatory T cells, along with the production of suppressive cytokines and metabolites, could lead to tumor progression [12]. Neoantigen-specific lymphocytes have not only been identified at the tumor site but also in the peripheral blood of melanoma patients [23]. A relocation to the metastatic sites after stimulation by an immune checkpoint blockade could be crucial for the antitumor response. The transfer to the tumor might be impaired by T-cell inhibition and consecutive lymphopenia in peripheral blood. In this case, lymphopenia is the surrogate marker for melanoma resistance to ICI, and treatment has to be adjusted to overcome resistance.

In summary, additional translational studies are needed to investigate the mechanism of lymphopenia and to improve the outcomes of patients who develop lymphopenia during ICI therapy. 

## 5. Conclusions

Lymphopenia is regularly observed in patients with melanoma treated with ICI. The detection of lymphopenia after the beginning of ICI therapy is a predictive marker and is associated with a significantly reduced PFS and OS. 

## Figures and Tables

**Figure 1 cancers-14-03282-f001:**
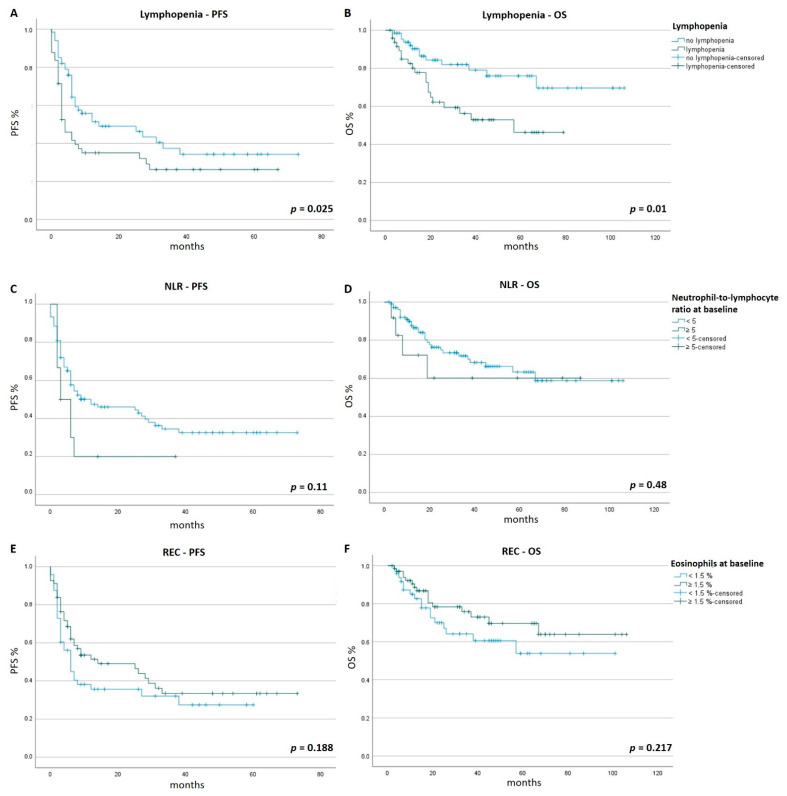
PFS and OS. Patients were followed for a minimum of 6 months. Tick marks indicate censored data. Time is given in months. Panels (**A**,**C**,**E**) show the Kaplan–Meier estimates of PFS of patients. Panels (**B**,**D**,**F**) show the Kaplan–Meier estimates of OS. Panel (**A**): the mean PFS of patients who developed lymphopenia was significantly shorter (13.3 months) compared to patients who did not develop lymphopenia (16.9 months, *p* = 0.025). Panel (**B**): the mean OS of patients who developed lymphopenia was significantly shorter (28.1 months) compared to patients who did not develop lymphopenia (36.8 months, *p* = 0.01). Panel (**C**): the mean PFS of patients with a NLR ≥ 5 at baseline was longer (16.3 months) compared to patients with a NLR < 5 at baseline (7.3 months, *p* = 0.11). Panel (**D**): the mean OS of patients with a NLR ≥ 5 at baseline was longer (33.6 months) compared to patients with a NLR ratio < 5 at baseline (28.8 months, *p* = 0.48). Panel (**E**): the mean PFS of patients with a relative eosinophilic count (REC) ≥ 1.5% at baseline was longer (16.9 months) compared to patients with a REC < 1.5% at baseline (13.3 months, *p* = 0.188). Panel (**F**): the mean OS of patients with a REC ≥ 1.5% at baseline was longer (34.2 months) compared to patients with a REC < 1.5% at baseline (31.6 months, *p* = 0.217).

**Table 1 cancers-14-03282-t001:** Baseline characteristics of the patients. Percentages are given in parentheses.

	Lymphopenia (*n* = 49)	No Lymphopenia (*n* = 67)	*p*-Values
**Age–y**			
Mean	60.6	62.9	0.350
Range	37–81	22–90	
**Sex–no. (%)**			
Male	34 (69.4)	31 (46.3)	**0.015**
Female	15 (30.6)	36 (53.7)	**0.015**
**ECOG performance status–no. (%)**			
0	47 (95.9)	62 (92.5)	0.697
1	2 (4.1)	5 (7.5)	0.697
**M stage–no. (%)**			
M1a	7 (14.3)	10 (14.9)	1.000
M1b	7 (14.3)	19 (28.4)	0.114
M1c	14 (28.6)	19 (28.4)	1.000
M1d	21 (42.9)	19 (28.4)	0.117
**Lactate dehydrogenase–no. (%)**			
≤ULN	37 (75.5)	54 (80.6)	0.648
>ULN	10 (20.4)	11 (16.4)	0.630
≥2 × ULN	2 (4.1)	2 (3.0)	1.000
**BRAF status–no. (%)**			
Mutation	25 (51.0)	22 (32.8)	0.057
No mutation	24 (49.0)	45 (67.2)	0.057
**NRAS status–no. (%)**			
Mutation	10 (20.4)	12 (17.9)	0.812
No mutation	39 (79.6)	55 (82.1)	0.812
**Treatment–no. (%)**			
Ipilimumab	5 (10.2)	4 (6.0)	0.490
Nivolumab	7 (14.3)	15 (22.4)	0.341
Pembrolizumab	6 (12.2)	20 (29.9)	**0.026**
Ipilimumab + Nivolumab	31 (63.3)	28 (41.8)	**0.025**

**Table 2 cancers-14-03282-t002:** PFS and OS are shown for patients who developed lymphopenia depending on the time-point of occurrence of lymphopenia. Early is defined as the onset of lymphopenia between the beginning of ICI therapy and up to 4 months after. Delayed is defined as the onset of lymphopenia between 4 and 12 months after the beginning of ICI therapy. Late is defined as the onset of lymphopenia more than 12 months after the beginning of ICI therapy. The three groups were compared with regard to PFS (*p* = 0.727) and OS (*p* = 0.878). ICI: immune checkpoint inhibitor.

	Early (0–4 Months) *n* = 16	Delayed (4–12 Months) *n* = 15	Late (>12 Months) *n* = 15	*p*-Values
**PFS** **(months)**	11.4 (range 0–50)	17.1 (range 0–67)	11.9 (range 0–44)	0.727
**OS** **(months)**	24.4 (range 2–62)	29.8 (range 3–70)	30.1 (range 3–66)	0.878

**Table 3 cancers-14-03282-t003:** PFS and OS, depending on antibiotic treatment and development of lymphopenia during ICI therapy. PFS and OS were compared between the different groups.

	No Antibiotic Treatment		Antibiotic Treatment		*p*-Values
**No Lymphopenia**	*n* = 59	PFS 17.2 months (range 1–73)	*n* = 8	PFS 14.8 months (range 2–54)	0.322
		OS 37.1 months (range 5–106)		OS 34.3 months (range 4–104)	0.512
**Lymphopenia**	*n* = 37	PFS 14.8 months (range 0–67)	*n* = 12	PFS 8.6 months (range 0–50)	0.976
		OS 26.6 months (range 2–70)		OS 33.2 months (range 3–79)	0.730

## Data Availability

Data are available on request from the authors.

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
