# Peer review of "Development of Lymphopenia during Therapy with Immune Checkpoint Inhibitors Is Associated with Poor Outcome in Metastatic Cutaneous Melanoma"

_cancers, 2022, doi:10.3390/cancers14133282_

Round 1

Reviewer 1 Report

In response to critiques, I believe the manuscript has been significantly improved and now warrants publication in Cancers.

Author Response

Thank you very much for reviewing our manuscript.

Reviewer 2 Report

Title of the article may be revised to indicate the authors' opinion more clearly. Is "at any time" necessary for the conclusion? It is not clear which groups were compared for calculating p-values in Table 2 and 3.

Conclusion may be revised to indicate more detailed settings of lymphopenia after the ICI therapy. 

Author Response

Title of the article may be revised to indicate the authors' opinion more clearly. Is "at any time" necessary for the conclusion?

We changed the title accordingly by omitting “at any time” in the title.

It is not clear which groups were compared for calculating p-values in Table 2 and 3.

In Table 2 PFS and OS were compared between the three different groups according to onset of lymphopenia as explained in the text (161 – 166). To make it clearer in the table 2, we added “the three groups were compared with regard of PFS (p = 0.727) and OS (p = 0.878)” in the legend.

To clarify in table 3 we accordingly added to the already existing explanation in the text (lines 179 – 198) “PFS and OS were compared between the different groups” to the legend.

            Conclusion may be revised to indicate more detailed settings of lymphopenia after the ICI therapy. 

We indicated the specific details for the findings in the discussion section:

"A statistically significant association was observed for both PFS and OS with worse outcome for patients who develop lymphopenia with PFS of 13.3 months in patients who develop lymphopenia compared to 16.9 months in patients without occurrence of lymphopenia. Patients with occurrence of lymphopenia had an OS of 28.1 months compared to 36.8 months in patients who did not develop lymphopenia. Differences in PFS and OS were observed in patients with early as well as late onset of lymphopenia after initiation of immunotherapy."

Reviewer 3 Report

I thank the authors for their work to address all comments. The authors provided adequate revision to their manuscript. The only thing I would suggest is to try to come up with deeper hypotheses (possible mechanism) for the observed phenomena in the Discussion section.

Author Response

Thank you for your feedback. We improved our discussion part to provide a deeper hypothesis:

"Our data underline the importance of circulating lymphocytes detectable in the blood count for anti-tumor response and protection against infections. The reason for the development of lymphopenia remains unclear but potentially includes (i) loss of lymphocytes as an immune-related hematological side effect [19], (ii) apoptosis triggered by Fas-ligand expressed within the tumor microenvironment and (iii) apoptosis due to excessive exhaustion. Lymphopenia as hematologic immune related adverse event, would be in line with the fact that the frequency was significantly higher in patients treated with combined immunotherapy with ipilimumab and nivolumab, which is known to be associated with higher rates of severe immune related adverse events compared to treatment with anti-PD1- or anti-CTLA-4 antibodies alone [3]. On the other hand, therapy with the anti CTLA-4 monoclonal antibody tremelimumab has been shown to restore CD4+ and CD8+ T-cells in 7 of 10 patients with pretreated advanced melanoma and severe lymphopenia [20].

The mechanism of apoptosis triggered by Fas-ligand was shown in a genetically engineered melanoma mouse model resistant to checkpoint blockade in which lymphocytes in the periphery were consecutively depleted [21]. Lastly, persistent antigen stimulation and immunosuppressive cytokines lead to T-cell dysfunction with elevated and sustained expression of inhibitory receptors [22], which is further intensified by ICI.

The reason for worse outcome in patients who develop lymphopenia could be due to changes in the tumor microenvironment. The accumulation of myeloid-derived suppressor cells, type-2 macrophages or regulatory T cells along with the production of suppressive cytokines and metabolites could lead to tumor progression [12]. Neoantigen-specific lymphocytes have not only been identified at the tumor site but also in the peripheral blood of melanoma patients [23]. A relocation to the metastatic sites after stimulation by immune checkpoint blockade could be crucial for the anti-tumor response. The transfer to the tumor might be impaired by T-cell inhibition and consecutive lymphopenia in peripheral blood. In this case lymphopenia is the surrogate marker for melanoma resistance to ICI and treatment has to be adjusted to overcome resistance.

In summary, additional translational studies are needed to investigate the mechanism of lymphopenia, and to improve the outcome of patients who develop lymphopenia during ICI therapy."

This manuscript is a resubmission of an earlier submission. The following is a list of the peer review reports and author responses from that submission.

Round 1

Reviewer 1 Report

The research manuscript titled "Development of lymphopenia at any time during therapy with immune checkpoint inhibitors is associated with poor outcome in metastatic cutaneous melanoma" by Tomsitz et al., is an interesting manuscript discussing the development of lymphopenia on the outcome of ICI on the clinical treatment of metastatic melanoma. 

The topic is interesting, and the manuscript is well-written. The results are clear and support the conclusions. The only concern is that Figure 2  must be professionally prepared. Otherwise, the manuscript is suitable for publication.

Reviewer 2 Report

The study describes that the patients with metastatic melanoma who develop lymphopenia during immune checkpoint inhibitor treatment have lower prognosis. Significance of the data may be indicated in the Figure 1 and   2.  It is not clear whether onset of lymphopenia has correlation with PFS or OS in the current Figure 2. The label for Y axis is needed. Table 2 is missing.

Reviewer 3 Report

This submitted article suggests as a potential biomarker for predicting outcomes after initiating immune chokepoint blockade ICB) in patients with metastatic cutaneous melanoma (mCM). While this approach may be a good addition to the field, the paucity of collected data and the hypothetical interpretation of these data do not support the main conclusion. Overall this is a very weak study with very speculative conclusions.

Although I understand that investigation of lymphocytes populations was not possible, I am not sure why the authors, despite having totally access to the CBC data with differential which made possible the study do not show the correlation between the rest of blood populations and the survival. At the end of the day even the authors mentioned in the paper the prognosis factor of eosinophils (line 61). Moreover neutrophils-to-lymphocytes ration is a much better prognosis hematologic parameter than lymphocyte alones.

An important table, patient’s co-morbidities and associated treatments which make impact lymphopenia, is not discussed/presented.

Cytokines measurement would have brought a humongous significance to the paper.

On what scientific premise Figure 2 classified onset of lymphopenia in early (0-4 months), delayed (4012 months) and late (after 12 months) time-points? Moreover, despite lacking the statically significance between apparently increment in OS, the PFS “came back” in late time-point as in early time-point suggesting that all these data are probably the consequence of random events, therefore not supporting the main hypothesis.

Had any patient in this investigation received any other immunosuppressive treatment as chemotherapy? This is not discussed.

A critical control group is missing; patients with mCM without ICB.

Not sure why but Table 2 is missing. Also, the table 2 should include the correlations based on type of antibiotic used and their route if administration. Were any blood or blood products, G-CSF, or bone marrow transplants administered to the patients in the study?

Sections of Discussion come from nowhere and do integrate well to the presented investigation (e.g., lines 216-218).

Table 1 does not mention what is described in the parenthesis (probably percentages)

Figure 2, occupying 1/3 of page, is irrelevant. A short table should have sufficed.

Finally, a main question remains. Did lymphopenia indeed predict the failure to ICB, or the cancer just progressed ICB-independent and impairing hematologic profiles? Therefore, all comprehensive labs should have been shown for a better correlation.